# Soil Anti-Scourabilities of Four Typical Herbaceous Plants and Their Responses to Soil Properties, Root Traits and Slope Position in Northeast China

Xueshan Wang [1], Mingming Guo [2], Jielin Liu [1,*], Xiaolei Kong [1], Daqing Peng [1] and Qiang Zhang [1]

[1] Department of Herbage Breeding, Prataculture Research Institute, Heilongjiang Academy of Agricultural Sciences, Harbin 150081, China
[2] Key Laboratory of Mollisols Agroecology, Northeast Institute of Geography and Agroecology, Chinese Academy of Sciences, Harbin 150081, China
* Correspondence: liujielin7857@163.com

**Abstract:** Vegetation has been proven to be an effective measure to mitigate soil erosion in most regions and climates. However, it is not clear how some herbaceous plants affect the ability of soil to resist slope flow erosion in the Mollisol region of Northeast China. In this study, four herbaceous plant plots of 50 m × 4.5 m, including *Zea mays* L., *Sorghum bicolor* × *Sudanense*, *Avena sativa* L. and *Lolium multiflorum* Lam., were established in a sloping land with an abandoned land as the control to detect the effect of herbaceous plants on soil anti-scourability (ANS). A hydraulic flume experiment was carried out to determine the soil ANS, and the root traits and soil properties were also measured at different slope positions. The results showed that the mean soil ANS ranged from 17.55 to 94.77 L g$^{-1}$ among different herbaceous plants, of which the *Lolium multiflorum* Lam. showed the strongest controlling effect on soil ANS (259.87%), followed by *Sorghum bicolor* × *Sudanense* (66.87%) and *Avena sativa* L. (18.12%), while the soil ANS of *Zea mays* L. decreased by 33.37% compared with the control. Soil ANS varied with slope position, and the mean soil ANS at the upslope was 116.50–134.21% higher than that of the middle slope and downslope. Additionally, soil ANS was positively related to root mass density (RMD), root length density (RLD), root surface area density (RSAD), soil total porosity and field capacity but was negatively related to soil bulk density ($p < 0.05$). Furthermore, the *Lolium multiflorum* Lam. exhibited better root distribution (i.e., high RSAD, RLD, RMD, and low root diameter) and soil physical structure (i.e., high soil porosity structure, water-holding capacity and low bulk density) than other plant species. Thus, the *Lolium multiflorum* Lam. is beneficial for enhancing soil erosion resistance to overland flow, especially at the up and middle slopes, and it could be preferred to control sloped soil erosion in Northeast China.

**Keywords:** slope erosion; soil anti-scourability; herbaceous plants; root traits; mollisols

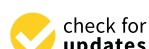



## 1. Introduction

Soil erosion is a global problem that severely threatens agricultural production and the ecological environment [1,2]. It has caused surface fragmentation, vegetation degradation and soil loss and altered the regional geographic environment by changing the migration and deposition patterns of soil particles [3–5]. Overland flow is the main driving force of soil erosion, and soil anti-scourability (ANS) is an effective index that indicates the ability of soil to resist scouring by runoff [6,7]. During the soil erosion process, slopes are predominantly erosional landforms [8], which occur mainly in sloping farmland and are generally considered the source of soil erosion. Controlling the material source of soil erosion has been proven to be a preferred measure for risk mitigation [9]. Consequently, it is crucial to evaluate soil erosion resistance and optimize soil conservation practices in sloping farmland, which is helpful for agricultural sustainability [10,11].

Previous studies have shown that vegetation plays an important role in controlling the soil erosion process by affecting land cover, root traits and soil properties [2,12–14] and improving soil resistance to rainfall or flowing water [15]. As reported by Wang et al. [16], the effect of vegetation on soil erosion mainly depends on vegetation types and vegetation density. Generally, grassland can significantly enhance soil ANS compared with cropland due to its developed root traits and positive effect on physiochemical soil properties [17,18]. For instance, Gyssels et al. [18] and Zhang et al. [14] reported that grasses are better at enhancing soil erosion resistance than cereals and shrub communities because their dense, fine roots can effectively capture runoff and sediment [16,19]. Xu et al. [20] also suggested that perennial herbs have great potential to weaken soil erosion by increasing soil ANS. Meanwhile, many studies have shown that herbaceous species are helpful in reducing raindrop splash and sediment in severely eroded areas [21,22]. However, there were significant differences in soil ANS among different herbaceous plants with different plant species and root types [20,22].

The root system, as a fundamental component of vegetation, can promote soil stability (e.g., soil aggregate stability, soil cohesion, and soil infiltration) by physical binding and chemical exudate-bonding effects [5,16,23,24] and hence enhance the soil resistance to erosion. Many studies have suggested that roots play a stronger role in preventing soil erosion than aboveground parts [25,26]. Wang et al. [27] found that more than half of the reduction in vegetation on soil erosion derives from living roots in grasslands. As plants grow, the root interweaves into the soil mass and reduces the impact of flowing water on soil detachment [18,28]. However, the effects of plant root systems on soil erosion vary with plant species and root morphology [15,16]. Several morphological traits, including root length density (RLD), root diameter (RD), root surface area density (RSAD) and root volume density, are used to characterize the effect of plant roots on soil erosion [15]. Previous publications indicated that soil ANS was significantly and positively related to root traits (i.e., root mass density (RMD), RLD, RSAD and RVD) [22,23,29,30]. Generally, RSAD is a primary root trait affecting soil erosion resistance because a larger contact area between plant roots and the soil mass can improve root bonding effects [16]. Moreover, fibrous roots showed greater potential to control soil erosion than tap roots because of the presence of numerous fine roots [15,18]. Guo et al. reported that roots less than 5 mm in diameter showed a more positive effect on soil loss than those with larger diameters [13].

In addition, soil properties also play an important role in mitigating soil erosion, while the growth of roots typically improves soil structure [5,31]. Many studies have demonstrated that soil bulk density (BD) can increase soil resistance to detachment by flowing water by enhancing soil strength and soil cohesion [32]. Wang et al. [16] reported that *L. secalinus* and *A. capillaris* grasslands with low soil BD and cohesion had high soil detachment capacities. However, some studies have also suggested that high soil BD can limit the development of roots by reducing root elongation and root density [33]. Chen et al. [31] found that soil with low BD was difficult to detach because of the greater root physical and soil organism activities. Meanwhile, plant roots showed a positive effect on the soil porosity structure, which caused better infiltration capacity and less surface runoff [16,19]. The improvement in soil porosity also enhances the soil water holding and absorption capacity [26].

The Mollisol region of Northeast China is essential for ensuring national food security [34,35]. Long-term intensive agricultural practices intensify soil erosion and seriously threaten the sustainability of local agricultural production and the ecological environment [5]. The major topographic features in the Chinese Mollisol region are characterized by long slope lengths and gentle slope gradients, which create topographic conditions for hillslope erosion by influencing soil water infiltration and runoff generation [36,37], and surface runoff is the main external erosive force of hillslope soil erosion in the Mollisol region [38]. Thus, some agronomic measures should be adopted to enhance soil resistance to flowing water in the Mollisol region [10,39]. In this study, four herbaceous plants were established on typical sloping land, and a hydraulic flume experiment was carried out

to (1) clarify the differences in soil ANS, soil properties, and root traits among different herbaceous plants, (2) determine the main factors affecting soil ANS, and (3) select a suitable plant species for slope erosion control in the Mollisol region of Northeast China. The purpose of this study is to provide an effective agronomic measure for controlling hillslope erosion and reducing runoff from agriculture in Northeast China.

## 2. Materials and Methods

### 2.1. Study Area

This study was conducted in the Guangrong watershed (1.86 km², Hailun County, 47.21° N, 126.50° E), which is located in the middle of the Mollisol region of Northeast China (Figure 1). The area has a continental monsoon climate with a mean annual temperature of 1.5 °C. The mean annual precipitation is 530 mm, with 65% distributed from June to August [40]. Rainfall events less than 25 mm h$^{-1}$ occupied approximately 95.6% of the year. The soil is classified as Mollisol (USST) with a silty clay-loam texture. More than 80% of the land is sloping farmland, and the terrain is characterized by rolling hills with long slope lengths (>200 m) and gentle slope gradients (<5°) in most areas. The soil erosion rate is 1.2–2.4 mm per year [10]. The main land use type is cropland, with more than 100 years of cultivation history, and maize (*Zea mays* L.) and soybean (*Glycine max*) are the primary crops [41].

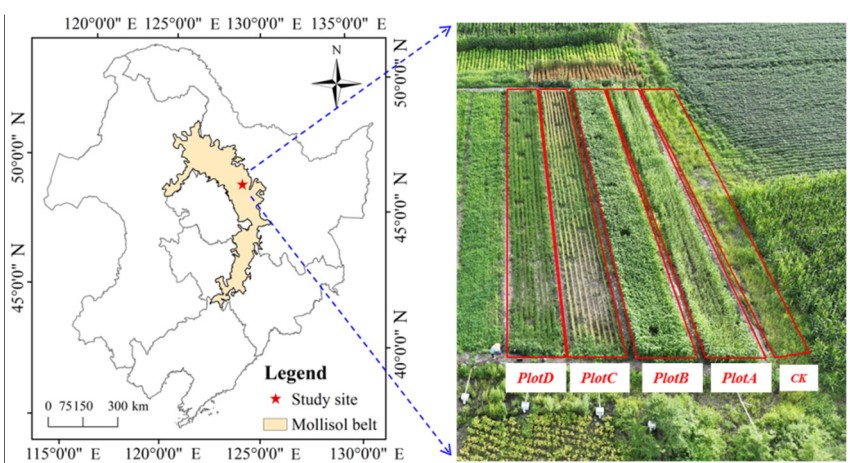

**Figure 1.** Location of the study area in the Mollisol region and aerial view of the experimental plots. Note: CK: abandoned land (*Echinochloa crusgalli (L.) Beauv*). PlotA: *Zea mays* L. PlotB: *Sorghum bicolor × Sudanense*. PLotC: *Avena sativa* L. PlotD: *Lolium multiflorum* Lam.

### 2.2. Design and Layout of Experimental Plots

Four slope runoff plots were established on sloping land in April 2022 (Figure 1). Each plot was 50 m long and 4.5 m wide and was separated from other plots by a 0.5 m buffer strip. The sloping land was an east-facing slope with downslope tillage practices. Four types of herbaceous plants, including Maize (*Zea mays* L.), Sorghum hybrid sudangrass (*Sorghum bicolor × Sudanense*), Oat (*Avena sativa* L.) and Ryegrass (*Lolium multiflorum* Lam.), were planted in May with normal fertilization and harvested in October. The four annual herbaceous plants belong to the *Poaceae Barnhart* and are annual plants, among which *Zea mays* L. is the main crop type in the study area, and the others showed high economic benefits according to the previous cultivar comparison test. Moreover, an abandoned land (the dominant species was *Echinochloa crusgalli (L.) Beauv*) adjacent to the experimental plots was chosen as a control plot. The background values of the soil's physicochemical properties are listed in Table 1.

**Table 1.** The background values of soil physicochemical properties in this study.

| Slope Position | Bulk Density (g cm$^{-3}$) | Total Porosity (%) | Saturated Moisture (%) | Field Capacity (%) | Soil Organic Carbon (g kg$^{-1}$) | Total Nitrogen (g kg$^{-1}$) | Total Phosphorus (g kg$^{-1}$) | pH |
|---|---|---|---|---|---|---|---|---|
| Up | 1.46 ± 0.13 | 32.98 ± 1.75 | 22.54 ± 3.36 | 19.11 ± 2.90 | 71.03 ± 8.03 | 2.53 ± 0.49 | 0.73 ± 0.05 | 6.00 ± 0.29 |
| Middle | 1.30 ± 0.05 | 40.77 ± 0.83 | 31.56 ± 1.60 | 22.97 ± 1.69 | 55.18 ± 9.31 | 1.66 ± 0.28 | 0.54 ± 0.01 | 6.11 ± 0.42 |
| Down | 1.38 ± 0.14 | 32.80 ± 6.57 | 24.31 ± 8.29 | 21.30 ± 5.10 | 43.83 ± 5.11 | 1.41 ± 0.28 | 0.46 ± 0.04 | 6.50 ± 0.53 |

*2.3. Soil Sampling and Measurement*

A total of 45 undisturbed topsoil samples, including five herbaceous plants, three slope positions and three replications, were collected randomly using a rectangular metal box (with a length of 25 cm, a width of 10 cm, and a height of 10 cm) for measuring soil ANS in August 2022. The aboveground part of the collected samples was cut off, followed by litter removal. Meanwhile, the soil around each sampling point was collected using a steel ring (50 mm in diameter, 50 mm in height) to determine the soil properties (i.e., soil BD, noncapillary porosity (NCP), total porosity (TP), moisture content (SM), field capacity (FC) and saturated moisture content (SSM)) by the oven-drying method [10], the detailed measure methods were referenced by Wang et al. [16].

*2.4. Hydraulic Flume Experiment*

The soil ANS was measured using a hydraulic flume that was 2.0 m long and 0.2 m wide. The width of the flume is higher than that of soil samples to avoid the boundary effect of the flume. The water supply device consists of a water supply tank, buffer tank, plastic pipe, flowmeter and valve. The device was adjusted repeatedly through a pretest to ensure a stable and uniform flow velocity [39]. Each soil sample was saturated using a watering pot before the experiment, which can eliminate the error caused by soil moisture (e.g., infiltration) during the measuring process [31]. The wetted sample was set into a rectangular opening and scoured under the desired flow discharge (40 L min$^{-1}$) and flume gradient (6°). Runoff and sediment samples were collected with sampling tanks, and the sampling time was recorded. The sampled sediment was oven-dried at 105 °C until constant weight to determine the soil loss amount (g). The measured soil loss amount was used to calculate the soil ANS using the following equation [6,22]:

$$\text{ANS} = \frac{\text{f} \times \text{T}}{\text{W}} \tag{1}$$

where ANS is the soil anti-scouring index per unit flow (L g$^{-1}$), f is the flow rate (L min$^{-1}$), T is the scouring time (min), and W is the oven-dried sediment weight (g).

Following the hydraulic flume experiment, the soil samples were repeatedly washed by the manual washing method to collect intact and clean roots [10]. Plant root traits, including RD, RLD and RSAD, were analyzed in WinRHIZO image analysis software (version 2007 pro). The washed roots were oven-dried for 48 h at 65 °C and weighed to determine root biomass.

*2.5. Statistical Analysis and Software*

One-way analysis of variance (ANOVA) was used to evaluate the differences in soil ANS, soil properties and root traits among the five herbaceous plants and three slope positions, and the significance level was set at 0.05 by the least-significant difference (LSD). The relationships between soil ANS and soil properties and root traits were identified by Pearson correlation analysis and regression analysis. A random forest model analysis was used to evaluate the relative importance of variables on soil ANS by the rfPermute package in Rstudio 4.1.0 (Crunchbase Inc., Boston, MA, USA). All statistical analyses were performed using SPSS 19.0 (IBM, Chicago, IL, USA) and Rstudio 4.1.0 (Crunchbase Inc., Boston, MA, USA), and figures were drawn by Origin 2017 (OriginLab, Northampton, MA, USA).

## 3. Results

### 3.1. Soil ANS of Different Herbaceous Plants

The mean values of soil ANS ranged from 17.55 to 94.77 L g$^{-1}$ for the five herbaceous plants (Figure 2a). Compared with abandoned land, the soil ANS decreased by 33.37% for conventional *Zea mays* L., while it increased by 66.87%, 18.12%, and 259.87% for *Sorghum bicolor × Sudanense*, *Avena sativa* L. and *Lolium multiflorum* Lam., respectively. No significant differences in soil ANS were found between the three slope positions ($p > 0.05$), but the upslope generally exhibited higher soil ANS (67.88 L g$^{-1}$), which was 116.50–134.21% higher than that of the middle slope and downslope. *Lolium multiflorum* Lam. showed the maximum soil ANS at the upslope (182.70 L g$^{-1}$) and middle slope (80.85 L g$^{-1}$), which were 2.80–7.18 and 2.87–9.79 times greater than the others, respectively. *Sorghum bicolor × Sudanense* had the highest soil ANS (43.48 L g$^{-1}$) at the downslope, and the conventional *Zea mays* L. generally showed a low soil ANS at different slope positions (8.26–32.55 L g$^{-1}$; Figure 2b).

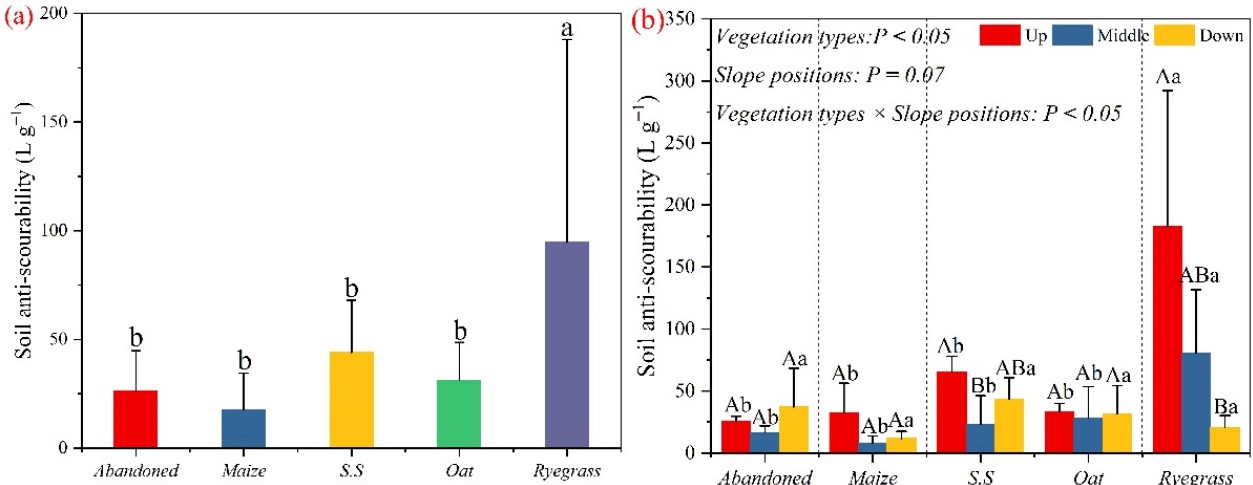

**Figure 2.** The soil ANS of different herbaceous plants (**a**) and different slope positions (**b**). Note: Abandoned: *Echinochloa crusgalli (L.) Beauv*. Maize: *Zea mays* L. S.S: *Sorghum bicolor × Sudanense*. Oat: *Avena sativa* L. Ryegrass: *Lolium multiflorum* Lam. Different lowercase letters indicate a significant difference among different herbaceous plants, and the different capital letters indicate differences among different slope positions at the $p < 0.05$ level, respectively.

### 3.2. Root Traits of Different Vegetation Types

Significant differences in root traits were found among different herbaceous plants ($p < 0.05$). The measured root diameter ranged from 0.36 to 0.65 mm, and conventional *Zea mays* L. had the highest root diameter, followed by *Sorghum bicolor × Sudanense*, *Echinochloa crusgalli (L.) Beauv*, *Lolium multiflorum* Lam. and *Avena sativa* L. (Figure 3a). The mean RLD ranged from 0.87 to 8.40 cm cm$^{-3}$, and the maximum found in *Lolium multiflorum* Lam. was 5.03 to 9.65 times greater than others ($p < 0.05$; Figure 3c). *Lolium multiflorum* Lam. also showed the highest RSAD (1.03 cm$^2$ cm$^{-3}$), which was 3.56 to 9.67 times greater than that of the other species ($p < 0.05$; Figure 3e). The RMD ranged from 0.19 to 1.80 kg cm$^{-3}$, and *Sorghum bicolor × Sudanense* exhibited the highest value (Figure 3f). Although no significant differences in root traits were found among the three slope positions, the ANOVA results showed that the middle slope had the highest root diameter (0.52 mm; Figure 3b), but its mean RLD was lowest (2.38 cm cm$^{-3}$; Figure 3d). The RSAD and RMD decreased along the slope position (Figure 3f,h).

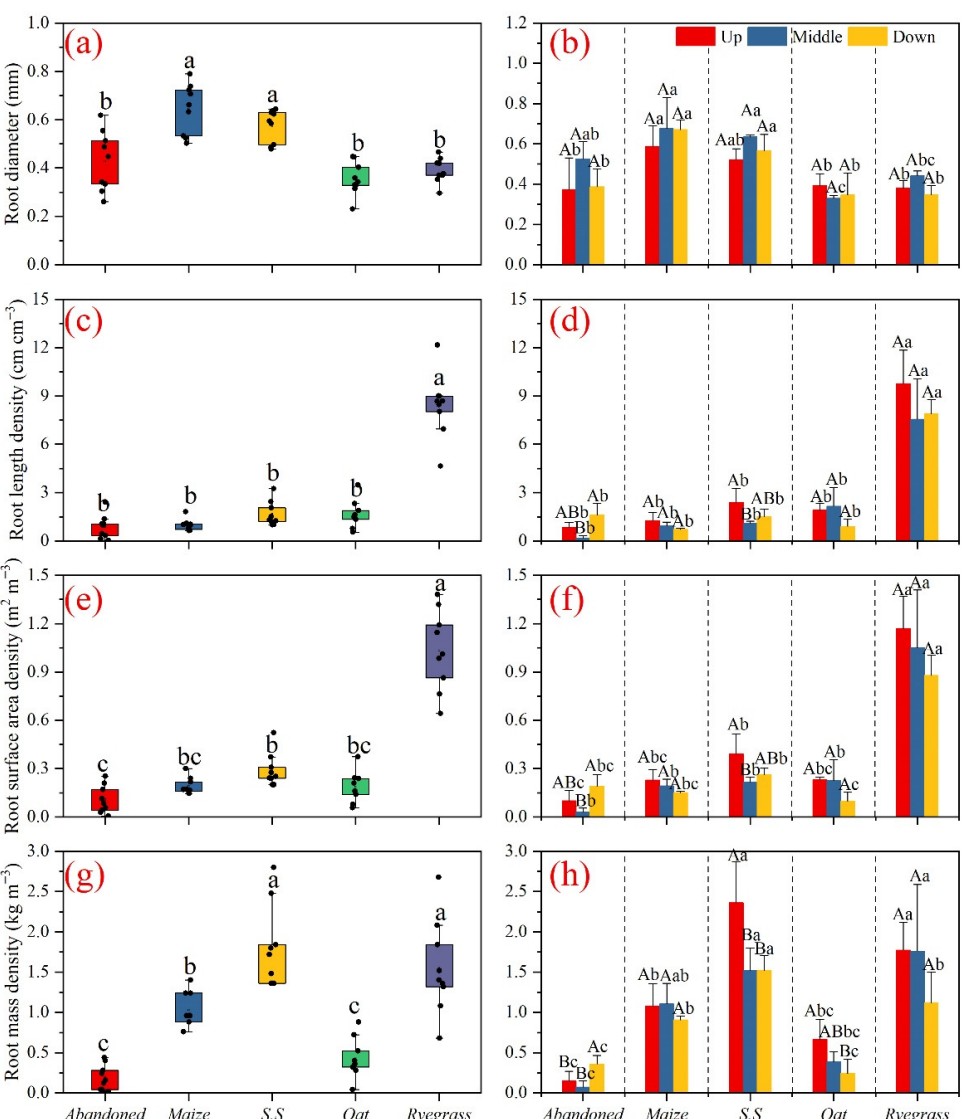

**Figure 3.** The root traits of different herbaceous plants (**a,c,e,g**) and different slope positions (**b,d,f,h**). Note: Abandoned: *Echinochloa crusgalli (L.) Beauv*. Maize: *Zea mays* L. S.S: *Sorghum bicolor × Sudanense*. Oat: *Avena sativa* L. Ryegrass: *Lolium multiflorum* Lam. Different lowercase letters indicate a significant difference among different herbaceous plants, and the different capital letters indicate differences among different slope positions at the *p* < 0.05 level, respectively.

### 3.3. Soil Properties of Different Vegetation Types

Soil properties varied significantly among the five herbaceous plants, while no differences were found between the three slope positions. The mean BD ranged from 1.32 to 1.49 g cm$^{-3}$, and the maximum (*Echinochloa crusgalli (L.) Beauv*) was 12.76% higher than the minimum (*Lolium multiflorum* Lam.; Figure 4a). The soil NCP and TP ranged from 3.83% to 8.67% and 37.54% to 44.04%, respectively, and *Lolium multiflorum* Lam. showed the best soil pore structure, followed by *Sorghum bicolor × Sudanense, Avena sativa* L., *Zea mays* L. and *Echinochloa crusgalli* (L.) Beauv (Figure 4c,e). Meanwhile, high FC (27.30%) and SSM (34.21%) were measured in *Lolium multiflorum* Lam., which were 0.87%–20.81% and 3.41%–35.54% higher than others, respectively (Figure 4i,k). However, a low SM (17.86%) was found in *Lolium multiflorum* Lam., which was 15.00% lower than the maximum (*Zea mays* L.; Figure 4g). Furthermore, the soil porosity structure and soil moisture content generally increased with slope position from up to down, while BD showed the opposite trend (Figure 4).

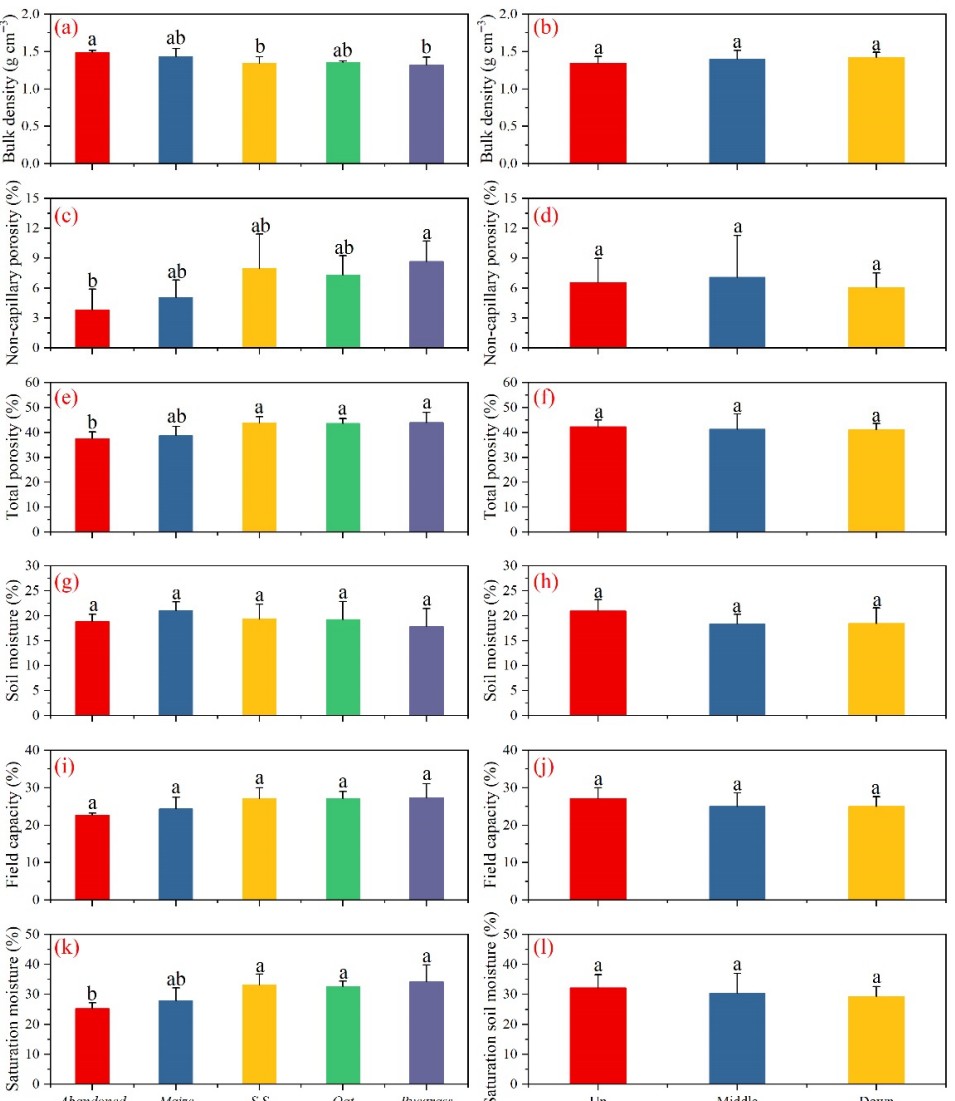

**Figure 4.** The soil properties of different herbaceous plants (**a**,**c**,**e**,**g**,**i**,**k**) and different slope positions (**b**,**d**,**f**,**h**,**j**,**l**). Note: Abandoned: *Echinochloa crusgalli (L.) Beauv*. Maize: *Zea mays* L. S.S: *Sorghum bicolor × Sudanense*. Oat: *Avena sativa* L. Ryegrass: *Lolium multiflorum* Lam. Different lowercase letters indicate a significant difference among different herbaceous plants and different slope positions at the *p* < 0.05 level, respectively.

### 3.4. Factors Affecting the Soil ANS

The RF results indicated that the root traits had a higher effect on the soil ANS than the soil properties. As shown in Figure 5, the sequences of relative importance were ranked as follows: RMD (6.97%; *p* < 0.01) > RSAD (5.90%; *p* < 0.05) > FC (5.84%; *p* < 0.05) > RLD (5.05%; *p* < 0.05) > BD (4.00%; *p* < 0.05) > TP (2.71%) > SSM (2.68%) > RD (2.06%) > SM (0.59%) > NCP (0.41%). Regression analysis indicated that the soil ANS increased with increasing RMD, RLD and RSAD with a linear function (*p* < 0.05). Regarding soil properties, the soil ANS increased with the increase in TP and FC with a power function but decreased with the increase in BD with a power function (*p* < 0.05; Figure 6). Meanwhile, the Pearson correlation results demonstrated that the soil ANS showed a significantly positive relationship with RSAD, RLD, RMD, soil TP, FC and SSM (*p* < 0.05). The soil ANS was negatively related to BD (*p* < 0.05) and RD (Figure 7).

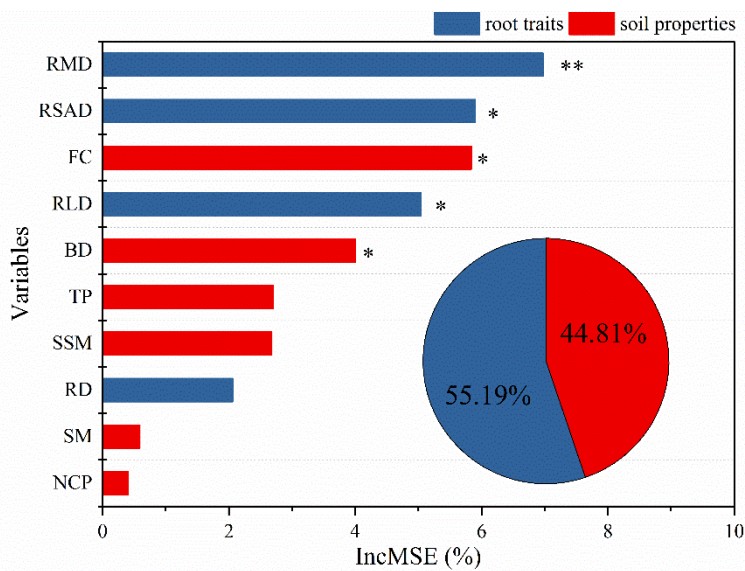

**Figure 5.** The relative importance of root traits and soil properties on soil ANS. Note: RMD: root mass density. RSAD: root surface area density. FC: field capacity. RLD: root length density. BD: soil bulk density. TP: total soil porosity. SSM: soil saturated moisture. RD: root diameter. SM: soil moisture content. NCP: noncapillary porosity. * represents the significance at the 0.05 level, ** represents the significance at the 0.01 level.

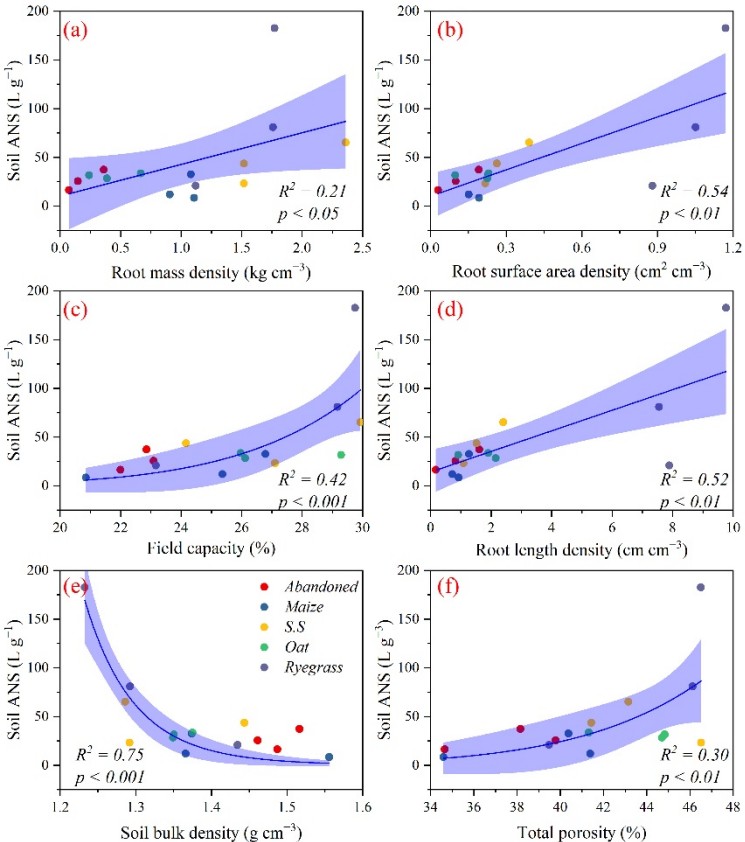

**Figure 6.** The relationships between soil ANS and root mass density (**a**), root surface area density (**b**), field capacity (**c**), root length density (**d**), soil bulk density (**e**) and soil total porosity (**f**). Note: Abandoned: *Echinochloa crusgalli (L.) Beauv*. Maize: *Zea mays* L. S.S: *Sorghum bicolor × Sudanense*. Oat: *Avena sativa* L. Ryegrass: *Lolium multiflorum* Lam.

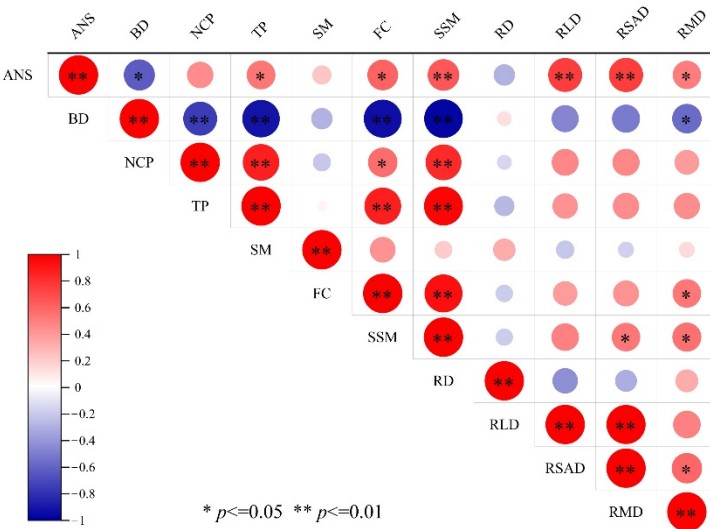

**Figure 7.** Correlation between soil ANS and root traits and soil properties. Note: RMD: root mass density. RSAD: root surface area density. FC: field capacity. RLD: root length density. BD: soil bulk density. TP: total soil porosity. SSM: soil saturated moisture. RD: root diameter. SM: soil moisture content. NCP: noncapillary porosity.

## 4. Discussion

### 4.1. Response of Soil ANS to Root Traits

Our results indicated that there was a significant difference in soil ANS among different herbaceous plants, and *Lolium multiflorum* Lam. exhibited the highest soil ANS (Figure 2a). This was mainly attributed to the differences in root systems among different herbaceous plants [15,16]. Many studies have pointed out that root systems can enhance the stability of soil's physical structure through their binding effect and bonding effect [27,42,43]. The downward penetration or lateral growth of roots can improve the soil's anti-shear capacity and tensile strength [19,28]. Meanwhile, root systems also play an important role in protecting soil against flow scouring by increasing soil roughness and infiltration capacity [13,44]. Generally, the effect of the root system on soil erosion depends on root biomass and root morphology (e.g., RD, RLD, and RSAD) [16,18,43]. Previous studies showed that soil detachment capacity decreased with increasing RMD, RLS, RSAD, and root area ratio [23,29,31]. In this study, we found that RSAD, RLD, and RMD were the main variables affecting soil ANS. This is consistent with the finding of Ma et al. in sloping farmland [30], and Sun et al. also reported that the soil detachment rate was more sensitive to RSAD and RLD [25].

Figure 3b showed that the mean RLD value of *Lolium multiflorum* Lam. was 5.03 to 9.65 times greater than that of the other species ($p < 0.05$), and the soil ANS was significantly positively related to RLD ($p < 0.05$; Figure 6d). This result was supported by some previous studies [22,23,45]. For example, Helsen et al. [46] reported that soil erosion resistance to the concentrated flow of herbaceous vegetation was highly related to RLD. High RLD values mean that more roots wedge into and wrap around the soil, which enhances the ability of the soil to withstand scouring by overland flows [24,45]. Meanwhile, a high RLD value was beneficial to the absorption of nutrients and water from the surrounding soil [22]. The results in this study indicated that the soil ANS also increased with increasing RSAD ($p < 0.05$; Figure 6b). Many studies have demonstrated that RSAD is the main root trait affecting soil erosion resistance because of the soil-fixing mechanism of the root system [15]. A high RSAD represented a large contact area between roots and soil as well as more rhizospheres, which enhanced the soil resistance to flowing water by increasing the root bonding effect [16,29]. Wang et al. [16] reported that RSAD could adequately reflect the effect of plant roots on soil erosion among different herbaceous species. Our results also showed that there was a significant difference in RSAD among different plant species, and

the mean value of *Lolium multiflorum* Lam. was 3.56 to 9.67 times greater than that of the others ($p < 0.05$; Figure 3e).

RMD is the most commonly used index to reflect the relationship between soil erodibility and plant roots, and their relationship is generally negative [15,39,47]. For instance, Guo et al. [2] reported that RMD was negatively related to the comprehensive soil erodibility index on the Loess Plateau. In this study, a positive relationship was found between soil ANS and RMD (Figure 6a), which was attributed to the positive direct effect of RMD on soil cohesion and water-stable aggregates [48]. Meanwhile, there was a significant difference in RMD among the different herbaceous species, and the RMD of the experimental plots was 2.23–9.31 times greater than that of the abandoned land ($p < 0.05$; Figure 3g). This may be due to the effect of tillage and fertilization on root growth. Moreover, previous studies indicated that no significant relationship was observed between soil ANS and RD [25,49], and our results confirmed this finding (Figure 7). However, it has been proven that the effect of fine roots on soil erosion is better than that of coarse roots [18,23]. Fine roots can enhance tensile strength and provide a larger interface between roots and soil [19,50]. In scouring tests, grass with well-developed fine roots was more effective in trapping sediment and runoff [51]. Many studies have suggested that roots with diameters of 0–1 mm have a greater effect on soil erosion [13,18]. Furthermore, root architecture also has a significant influence on soil erosion [24]. In this study, all herbaceous species had fibrous roots, and their mean RD ranged from 0.36 to 0.65 mm (Figure 3a). Generally, the herbaceous plants with well-developed and fine root systems showed satisfactory effects on mitigating soil erosion.

### 4.2. Response of Soil ANS to Soil Properties

Our results found that planting grasses significantly improved soil physical structure and hence increased the resistance of soil erosion to runoff (Figure 4). Generally, the penetration of root systems can enhance soil porosity and infiltration capacity [5], which consequently increases the ANS of soil mass by altering the vertical migration of soil moisture and the lateral transport of surface runoff [39,51]. In this study, the soil ANS was positively related to soil TP with a power function ($p < 0.05$; Figure 6f). A higher soil porosity structure was found in *Lolium multiflorum* Lam. with a dense root network (Figure 4e). *Lolium multiflorum* Lam. also had a higher SSM and FC, and a significantly positive relationship was found between soil ANS and SSM (Figure 7). Previous studies showed that the soil infiltration capacity increased with increasing soil TP and SSM [52], thus reducing overland flow and decreasing soil detachment. Meanwhile, root systems also improve water absorption and soil water holding capacity by capillary pores and vegetation transpiration [26]. Furthermore, many studies demonstrated that soil cohesion and soil strength increase with increasing soil BD, which results in the soil being hard to be scoured by overland flow [16,44]. However, the soil ANS was negatively related to the soil BD in this study, which was consistent with the findings of Wang et al. in the Loess Plateau [53]. This result can be attributed to the following: (1) the increasing soil BD limited the development of root systems and hence decreased soil ANS [33], and a significantly negative relationship between RMD and soil BD was found in this study (Figure 7); (2) the soil with low BD had greater root physical and soil organism activities [31]; and (3) soil BD significantly affected the soil infiltration capacity and penetration resistance [11]. All of these processes can change the soil's erosion resistance. In summary, *Lolium multiflorum* Lam. was suggested to enhance soil ANS by increasing soil porosity structure and water holding capacity and decreasing soil BD.

### 4.3. Response of Soil ANS to Slope Position

The soil ANS, root traits and soil properties varied with slope positions, and the upslope generally exhibited developed root systems, superior soil physical structure and a strong ability to resist soil erosion in this study. The mean ANS of the upslope was 116.50–134.21% higher than that of the middle slope and downslope (Figure 2). Previ-

ous studies also indicated that the soil erosion rate varied with slope position because of the difference in erosive forces and consequently affected the vegetation characteristics and soil properties [2,38]. Generally, the erosion-deposition patterns induced by water erosion showed slight erosion on the upper slope and severe erosion on the middle and lower slopes [37,38]. Similarly, the soil ANS, root traits and soil properties decreased with slope position in this study, but no significant differences were found among the three slope positions. This may be attributed to the following: (1) The selected sloping land is adjacent to the gully, long-term soil erosion caused many soil nutrients and sediment to be transported to gullies [41], and low soil nutrients were found at the downslope (Table 1), which limited the growth of vegetation [54]. (2) During the growth of vegetation, we observed large amounts of sediment and runoff deposited at the downslope, which further limited the growth of herbaceous plants due to the anaerobic environment [35]. Both of these processes resulted in a decrease in soil quality because poor vegetation coverage cannot effectively trap sediment and runoff in the rainy season [39]. The RMD of the downslope was 14.43% and 31.40% lower than that of the middle slope and upslope, respectively (Figure 3h). Moreover, the insignificant difference among slope positions may be due to the shorter slope length (50 m) in this study. In summary, we recommend planting *Lolium multiflorum* Lam. to enhance the soil ability against scouring by overland flow in Mollisol sloping land, especially at the upslope and middle slope. *Sorghum bicolor* × *Sudanense* with developed root systems may be an alternative at the downslope.

## 5. Conclusions

This study indicated that planting herbaceous plants on sloping land effectively increased the soil erosion resistance to flowing water by promoting root development and enhancing the soil's physical structure. *Lolium multiflorum* Lam. had the highest soil ANS, which was 259.87% greater than that of the control, followed by *Sorghum bicolor* × *Sudanense* and *Avena sativa* L., while *Zea mays* L. played a poor role in mitigating soil erosion. The upslope generally showed a 116.50–134.21% higher soil ANS than the middle slope and downslope. Both root traits (i.e., RSAD, RLD, and RMD) and soil properties (i.e., soil BD, TP, FC, and SSM) significantly affected the soil ANS. But root traits played a more important role in affecting soil ANS compared with soil properties. Our results recommended *Lolium multiflorum* Lam. as the main herbaceous plant to prevent hillslope erosion by improving soil physical structure and reducing runoff from agriculture in the Mollisol region of Northeast China.

**Author Contributions:** Conceptualization, X.W., M.G. and J.L.; methodology, X.W., M.G. and J.L.; software, X.W. and D.P.; validation, X.K. and Q.Z.; formal analysis, X.W. and X.K.; investigation, X.W., J.L. and D.P.; resources, X.W., J.L. and D.P.; data curation, X.W. and J.L.; writing—original draft preparation, X.W. and M.G.; writing—review and editing, J.L. and M.G.; visualization, X.W., D.P. and Q.Z.; supervision, J.L.; project administration, J.L.; funding acquisition, J.L. All authors have read and agreed to the published version of the manuscript.

**Funding:** This research was funded by the National Key Research and Development Program of China (grant number 2021YFD1500800) and the Modern Agro-industry Technology Research System (grant number CARS-34).

**Data Availability Statement:** Summarized data are presented and available in this manuscript, and the rest of the data used and/or analyzed are available from the corresponding author upon reasonable request.

**Conflicts of Interest:** The authors declare no conflict of interest.

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
