# Peer review of "Soil Anti-Scourabilities of Four Typical Herbaceous Plants and Their Responses to Soil Properties, Root Traits and Slope Position in Northeast China"

_sustainability, doi:10.3390/su142416807_

Round 1

Reviewer 1 Report

Manuscript ID: sustainability-2066183

An article entitled Soil anti-scourabilities of four typical herbaceous plants and their responses to soil properties, root traits and slope position in the Northeast China is in line witch the aim and scope of Suitability and may by published after minor revision. The more imported remarks are listed below:

1. Materials and Methods: Zea mays, Sorghum bicolor × Sudanense, Avena sativa L. and Lolium were chosen to the research, why the last has only generic name? The full two-part name of the species should be given.

2. There is lack of information why such plant species set were chose to the study?

3. What kind of land stabilization do the authors mean? In brownfield reclamation, or about reducing runoff from agriculture? - how would the selection of three grains for the experiment suggest? Why were three crops and one wild grass chosen? If the use of these species were to be reclamation, the entire study would be pointless and the results were predictable. It is well known that annual crops with a shallow and poorly developed root system will be of marginal importance in preventing erosion compared to, for example, long-standing grasses such as Lolium or long-standing shrubs (such as Salix) even better suited for this. These doubts should be answered both in the introduction and the conclusions.

Author Response

Dear Reviewer:

Thank you for your comments concerning our manuscript entitled “Soil anti-scourabilities of four typical herbaceous plants and their responses to soil properties, root traits and slope position in the Northeast China” (sustainability-2066183). We thank you for giving us the opportunity to revise the manuscript. These comments were valuable and helpful for revising and improving our paper. We have studied the comments carefully and made corresponding revisions and statements. Any revisions in the manuscript were marked up using the “Track Changes” function. The main corrections in the paper and the responses to the reviewers’ comments are as follows.

An article entitled Soil anti-scourabilities of four typical herbaceous plants and their responses to soil properties, root traits and slope position in the Northeast China is in line witch the aim and scope of Suitability and may by published after minor revision. The more imported remarks are listed below:

  1. Materials and Methods: Zea mays, Sorghum bicolor × Sudanense, Avena sativa L. and Lolium were chosen to the research, why the last has only generic name? The full two-part name of the species should be given.

Response: Accepted. We have supplemented the species name and checked in the manuscript. Please see line 128-130.

  1. There is lack of information why such plant species set were chose to the study?

Response: Thank you for your suggestion. The four annual herbaceous plants belong to the Poaceae Barnhart, among which Zea mays L. is the main crop type in the study area, and the others showed high economic benefits according to the previous cultivar comparison test in Northeast China, the information on plant species has been added in the manuscript. Please see line 131-134.

  1. What kind of land stabilization do the authors mean? In brownfield reclamation, or about reducing runoff from agriculture? - how would the selection of three grains for the experiment suggest? Why were three crops and one wild grass chosen? If the use of these species were to be reclamation, the entire study would be pointless and the results were predictable. It is well known that annual crops with a shallow and poorly developed root system will be of marginal importance in preventing erosion compared to, for example, long-standing grasses such as Lolium or long-standing shrubs (such as Salix) even better suited for this. These doubts should be answered both in the introduction and the conclusions.

Response: Thank you for your suggestion. The land stabilization mentioned in this study involves improving the soil physical structure and reducing runoff from agriculture. The relevant content was supplemented in the manuscript, please see line 104-106.

Indeed, as mentioned in the introduction (line 50-59), the erosion resistance of different vegetation types was significantly different, and Gyssels et al. [18] and Zhang et al. [14] reported that grasses are better at enhancing soil erosion resistance than cereals and shrub communities because their dense fine roots can effectively capture runoff and sediment [16,19]. The main objective of this study was also to compare the differences in soil ANS between different herbaceous plants. The four herbaceous plants selected in this study belong to the Poaceae Barnhart and are annual plants, and the above-ground parts were harvested in October. In addition, rainfall in this study area mainly distributed from June to August, and soil erosion was also mainly concentrated in this period. The collection of soil samples was carried out at the end of August, when the vegetation coverage of each herbaceous plant was higher. Thus, the high soil ANS of Lolium multiflorum Lam. is not caused by long standing. Some unclear states have been improved in the manuscript.

We tried our best to improve the manuscript and made some changes. These changes did not influence the content and framework of the paper.

Reviewer 2 Report

1. This study showed that both root traits and soil properties significantly affected the soil ANS, and recommended Lolium as the main herbaceous plant to prevent hillslope erosion driven by runoff in Mollison region of Northeast China. The content of this study is succinctly described. Moreover, its research design, hypotheses, methods and results are clearly stated. 

2. Using flume with scale of 2 m long and 0.2 m wide might result in boundary effect of flume and could not get uniform flow condition. It would be better if the authors could describe and analysis deeply in the content.

Author Response

Dear Reviewer:

Thank you for your comments concerning our manuscript entitled “Soil anti-scourabilities of four typical herbaceous plants and their responses to soil properties, root traits and slope position in the Northeast China” (sustainability-2066183). We thank you for giving us the opportunity to revise the manuscript. These comments were valuable and helpful for revising and improving our paper. We have studied the comments carefully and made corresponding revisions and statements. Any revisions in the manuscript were marked up using the “Track Changes” function. The main corrections in the paper and the responses to the reviewers’ comments are as follows.

  1. This study showed that both root traits and soil properties significantly affected the soil ANS, and recommended Lolium as the main herbaceous plant to prevent hillslope erosion driven by runoff in Mollison region of Northeast China. The content of this study is succinctly described. Moreover, its research design, hypotheses, methods and results are clearly stated.

Response: Thank you for your comment.

  1. Using flume with scale of 2 m long and 0.2 m wide might result in boundary effect of flume and could not get uniform flow condition. It would be better if the authors could describe and analysis deeply in the content.

Response: We fully agree with your opinion that the scale of flume might cause a boundary effect of flume. The width of the flume is higher than that of soil sample in this study, which can avoid the boundary effect of flume. Moreover, before the scouring experiment, we conducted a pretest to repeatedly adjust the experiment device to ensure the stable and uniform flow velocity according to Yan et al. [39]. The relevant content was supplemented in the manuscript. Please see line 150-158.

We tried our best to improve the manuscript and made some changes. These changes did not influence the content and framework of the paper.

Reviewer 3 Report

There are no reservations to the subject of the manuscript, which is suitable for the Sustainability Journal. Soil scouring is a global issue, which seriously affects agricultural production and the environment, that is why I believe the paper written by Xueshan Wang et al. is practical and offers important insights into agricultural practices, in particular in Northeast China.

There are no major reservations to the abstract. It contains information about the conducted experiment and its results.

Introduction:

The introduction does not clearly state the purpose of the research – please amend.

Materials and Methods

There are no reservations to this section. However, it is recommended that a study for the purpose of agricultural practice lasts for at least two years.

Result

There are no major reservations to the results. Please correct Figure 2 (b) – it is illegible. Line 233 – error  Fig. (2f, 2h), should be Fig. 3f, 3h). Please revise and correct that.

Discussion

The Discussion is well structured, the research results are confronted with other authors’ research. The editing requirements need to be adhered to in the entire text. It says fig., should be Figure. There are also editing errors in table and figure captions. E.g. it says Table 1, should be Table 1. The same applies to figure captions.  I suggest merging the Results with the Discussion.

Conclusions

Please expand on this part.

References

All references fail to comply with the referencing style guide of the journal – please correct them.

Subject to minor additions, the paper can be published in the Sustainability Journal.

Author Response

Dear Reviewer:

Thank you for your comments concerning our manuscript entitled “Soil anti-scourabilities of four typical herbaceous plants and their responses to soil properties, root traits and slope position in the Northeast China” (sustainability-2066183). We thank you for giving us the opportunity to revise the manuscript. These comments were valuable and helpful for revising and improving our paper. We have studied the comments carefully and made corresponding revisions and statements. Any revisions in the manuscript were marked up using the “Track Changes” function. The main corrections in the paper and the responses to the reviewers’ comments are as follows.

There are no reservations to the subject of the manuscript, which is suitable for the Sustainability Journal. Soil scouring is a global issue, which seriously affects agricultural production and the environment, that is why I believe the paper written by Xueshan Wang et al. is practical and offers important insights into agricultural practices, in particular in Northeast China.

There are no major reservations to the abstract. It contains information about the conducted experiment and its results.

Response: Thank you for your comment.

Introduction:

The introduction does not clearly state the purpose of the research – please amend.

Response: Accepted. We have clarified the purpose of this research in the introduction. Please see line 104-106.

Materials and Methods

There are no reservations to this section. However, it is recommended that a study for the purpose of agricultural practice lasts for at least two years.

Response: We fully agree with your opinion that the study about agricultural practices should be at least two years. Although it is the first year of this study, the rainfall characteristic of this year was highly representative of the average annual rainfall in this area. We believe that this study can reflect the effect of vegetation measures on the efficiency of water storage and sediment reduction, which can achieve our research purpose. Of course, the result of long-term research is more accurate as you mentioned, we will continue to observe the effects of the long-term herbaceous planting on soil erosion processes, and we look forward to your continued support and suggestions.

Result

There are no major reservations to the results. Please correct Figure 2 (b) – it is illegible. Line 233 – error Fig. (2f, 2h), should be Fig. 3f, 3h). Please revise and correct that.

Response: We are so sorry for the wrongs. These wrongs have been corrected in the manuscript. Please see line 217-227 and Figure 2b.

Discussion

The Discussion is well structured, the research results are confronted with other authors’ research. The editing requirements need to be adhered to in the entire text. It says fig., should be Figure. There are also editing errors in table and figure captions. E.g. it says Table 1, should be Table 1. The same applies to figure captions. I suggest merging the Results with the Discussion.

Response: Accepted. Any inconsistencies with the editing requirements have been corrected in the manuscript, please see line 121, 138, 208, 244, 263, 304, 329 and 360.

Moreover, the objective of this study was to clarify the differences in soil ANS, soil properties and root traits among different herbaceous plants and determine the main factors affecting soil ANS. Therefore, we divide the discussion into three parts, including the response of soil ANS to root traits, soil properties and slope position, respectively. We thought it might be clearer to write the discussion separately.

Conclusions

Please expand on this part.

Response: Accepted. The conclusions have been further improved and expanded. Please see line 391-402.

References

All references fail to comply with the referencing style guide of the journal – please correct them.

Response: We are so sorry for the wrong, we have modified the format of the references in the manuscript according to the referencing style guide of the journal.

Subject to minor additions, the paper can be published in the Sustainability Journal.

We tried our best to improve the manuscript and made some changes. These changes did not influence the content and framework of the paper.

Reviewer 4 Report

Dear Authors,

As you state in the first sentence of the abstract, vegetation is, in general, an important factor in soil protection against erosion. This means that the study undertaken is not remarkable for its originality, the results obtained being important only for farmers and decision-makers in the area where they were carried out.

Nevertheless, the experiment is well conducted and the obtained results are well interpreted and can constitute a model for similar investigations. Therefore, I consider that the manuscript can be published with the following recommendations:

1. Please make some clarifications on the rainfall intensity in the studied area.

2. Subpoints 2.4 and 2.5 must be reformulated because they represent overlaps from other articles.

Thank you

Author Response

Dear Reviewer:

Thank you for your comments concerning our manuscript entitled “Soil anti-scourabilities of four typical herbaceous plants and their responses to soil properties, root traits and slope position in the Northeast China” (sustainability-2066183). We thank you for giving us the opportunity to revise the manuscript. These comments were valuable and helpful for revising and improving our paper. We have studied the comments carefully and made corresponding revisions and statements. Any revisions in the manuscript were marked up using the “Track Changes” function. The main corrections in the paper and the responses to the reviewers’ comments are as follows.

As you state in the first sentence of the abstract, vegetation is, in general, an important factor in soil protection against erosion. This means that the study undertaken is not remarkable for its originality, the results obtained being important only for farmers and decision-makers in the area where they were carried out.

Nevertheless, the experiment is well conducted and the obtained results are well interpreted and can constitute a model for similar investigations. Therefore, I consider that the manuscript can be published with the following recommendations:

  1. Please make some clarifications on the rainfall intensity in the studied area.

Response: Thank you for your comment. The monitoring data during 2012-2017 showed that the rainfall events less than 25 mm h-1 occupied approximately 95.6% of the year. The relevant content was supplemented in the manuscript, please see line 113-114.

  1. Subpoints 2.4 and 2.5 must be reformulated because they represent overlaps from other articles.

Response: We are sorry for the overlaps in the subpoints 2.4 and 2.5, the experiment methods introduced in the manuscript mostly refer to the pervious articles of our team. And we also have integrated and improved the two parts in the manuscript, please see line 150-175.

We tried our best to improve the manuscript and made some changes. These changes did not influence the content and framework of the paper.